# A New RNA-Dependent Cas12g Nuclease

**DOI:** 10.3390/ijms242317105

**Published:** 2023-12-04

**Authors:** Natalia Gunitseva, Martha Evteeva, Aleksei Korzhenkov, Maxim Patrushev

**Affiliations:** Complex of NBICS Technologies, National Research Center “Kurchatov Institute”, 123182 Moscow, Russiapatrushev_mv@nrcki.ru (M.P.)

**Keywords:** Cas12g, CRISPR/Cas-based detection, RNA-targeting systems, Cas proteins, *cis*-cleavage, *trans*-cleavage

## Abstract

The development of RNA-targeting CRISPR-Cas systems represents a major step forward in the field of gene editing and regulation. RNA editing presents a viable alternative to genome editing in certain scenarios as it offers a reversible and manageable approach, reducing the likelihood of runaway mutant variants. One of the most promising applications is in the treatment of genetic disorders caused by mutations in RNA molecules. In this study, we investigate a previously undescribed Cas12g nuclease which was found in metagenomes from promising thermophilic microbial communities during the expedition to the Republic of North Ossetia—Alania in 2020. The method outlined in this study can be applied to other Cas orthologs and variants, leading to a better understanding of the CRISPR-Cas system and its enzymatic activities. The cis-cleavage activity of the new type V-G Cas effector was indicated by in vitro RNA cleavage experiments. While CRISPR-Cas systems are known for their high specificity, there is still a risk of unintended cleavage of nontargeted RNA molecules. Ultimately, the search for new genome editing tools and the study of their properties will remove barriers to research in this area. With continued research and development, we may be able to unlock their full potential.

## 1. Introduction

The CRISPR-Cas system is a defense mechanism of the adaptive immune system, which is widespread among archaea and bacteria that protect them against viruses and other foreign genetic material. It uses Cas proteins to identify and cut specific nucleic acid sequences that match the guide RNA [1,2,3,4,5]. This immune system is exceptionally diverse into two classes, six types (I–VI), and 46 subtypes, in line with the recent classification [6,7]. The class 1 system (types I, III, and IV) employs multi-protein effector complexes to cleave foreign nucleic acids, while the class 2 system (types II, V, and VI) utilizes a single multi-domain effector endonuclease [8,9]. The identification of Cas9 RNA-guided endonuclease function initiated a surge of biotechnological advancements as scientists gained a better understanding of the enzymatic activities of CRISPR Cas enzymes [10,11]. The mechanism of action of the immunity process provided by the CRISPR-Cas system includes three main stages: adaptation, expression, and interference [12]. The CRISPR array and *cas* genes, which form one or more operons, are key components of this system. The CRISPR array is composed of direct repeats separated by spacers, which are fragments of non-native viral or plasmid DNA. This enables cells to effectively identify and eliminate previously encountered pathogens [13]. Genome editing technology based on the CRISPR-Cas system includes two mandatory components: a guide RNA and an effector (a protein or protein complex) that cleaves the target nucleic acid.

The CRISPR-Cas system is a widely utilized tool in various scientific fields, including molecular biology, medicine, and biotechnology. Its impact on these industries is profound, as it enables precise genome editing and has numerous applications in both experimental and applied research [14]. CRISPR-Cas class 2 genome editing systems are of particular interest to scientists because they contain one effector, which allows them to be conveniently used as tools for genomic engineering and nucleic acid detection [15,16,17,18]. Class 2 CRISPR-Cas systems include types II, V, and VI, with corresponding effectors Cas9, Cas12, and Cas13 [19,20]. Type V CRISPR-Cas systems vary by a single Cas12 effector that contains a conserved RuvC-like endonuclease domain regulated by RNA [21]. Typically, these effectors utilize this domain to cleave DNA substrates. However, Cas12g, the Cas effector of the recently discovered type V-G systems, distinguishes itself by displaying RNA-guided ribonuclease (RNase) activity instead of DNA cleavage. Also, Cas12g demonstrates collateral activity towards RNA and ssDNA once activated by RNA-targeting [22]. RNA editing presents a viable alternative to genome editing in certain scenarios as it offers a reversible and manageable approach, reducing the likelihood of runaway mutant variants. Unlike DNA editing, RNA recognition is not impeded by DNA modifications such as chromatin accessibility. This has led to a surge in the use of RNA-editing technologies, which offer numerous capabilities [23].

The CRISPR-Cas RNA-targeting systems enable specific targeting of nucleic acid fragments, including RNA molecules, and can cleave RNA upon target detection. One of the most promising applications of RNA-targeting CRISPR-Cas systems is in the treatment of genetic disorders caused by mutations in RNA molecules. For example, Huntington’s disease is caused by the accumulation of a mutant form of the huntingtin protein, which is encoded by an RNA molecule [24]. Moreover, the development of CRISPR-Cas RNA-targeting systems with dead nuclease activity (dCas) allows for further modification of this activity [25]. Overall, the development of RNA-targeting CRISPR-Cas systems represents a major step forward in the field of gene editing and regulation. In addition, the Cas12g protein is relatively small, consisting of only about 768 amino acids. This compact size makes it well suited for being inserted into therapeutic viral vectors like adeno-associated virus (AAV). While there are still challenges that need to be overcome, continued research and development in this area could lead to new discoveries and innovations in biotechnology and medicine. With the potential to revolutionize the way we treat genetic disorders and infectious diseases, RNA-targeting CRISPR-Cas systems hold immense promise for the future of healthcare.

Despite the vast knowledge of nucleases, there is still much to discover about their diversity and functions. Metagenomics has emerged as a powerful tool to explore microbial diversity and uncover new genes and enzymes. In particular, metagenomics has been used to identify novel nucleases from diverse habitats. The search for new nucleases in metagenomics data is important for several reasons. First, it can lead to the discovery of enzymes with unique properties, such as higher specificity or activity under extreme conditions. Second, the discovery of new nucleases can contribute to the development of new strategies for combating infectious diseases and antibiotic resistance.

Scientists all over the world are primarily focused on the research and development of genome editing, particularly with the utilization of CRISPR-Cas tools. The study of novel nucleases continues to be a major pursuit in molecular biology, as a single tool that can be applied in all areas has not yet been discovered. One such area of focus is the identification and examination of new subtypes of class 2 systems, specifically type V systems, in order to expand the CRISPR-Cas toolbox. In this study, we investigate a previously undescribed Cas12g nuclease, which was found in metagenomes from promising thermophilic microbial communities during the expedition to the Republic of North Ossetia—Alania in 2020. The method outlined in this study can be applied to other Cas orthologs and variants, leading to a better understanding of the CRISPR Cas system and its activities.

## 2. Results and Discussion

### 2.1. Identification and In Silico Characterization of Cas12g Nuclease

The sequence analysis showed that the Cas12g protein belongs to the CRISPR-Cas type V superfamily of enzymes that contain a RuvC nuclease domain that is responsible for RNA-guided cleavage. This recently discovered endonuclease displays collateral RNase and single-strand DNase activities upon activation by target molecular RNA, similar to Cas12a and Cas13 nucleases. Its bilobed architecture consists of a recognition lobe (REC) and a nuclease lobe (NUC) [9,22,26,27,28] (Figure 1B). To target ssRNA, the known Cas12g nucleases utilize two RNAs (a tracrRNA and a crRNA) without requiring a protospacer adjacent motif (PAM) sequence [29].

Candidate gene Cas12g was found in the metagenomic assembly of sample 4135 from the thermal springs of the Karmadon Gorge at an altitude of 2330 m above sea level. The microbial habitat was characterized by the temperature range of 52 to 55 °C, pH 6.1, and a weakly reducing potential from −20 to 0 mV. The aqueous phase was enriched with the following elements: K, Si, B, As, Fe, Li, Rb, Mn, Be, and Ba [30]. The Cas12g candidate gene most likely belongs to a member of the class *Terriglobia*; phylogenetic analysis shows that the evolutionary nearest homologs belong to members of *Terriglobia*, Planctomycetota, and several candidate divisions (Figure 1C). The CRISPR locus contained a *cas12g* gene capable of encoding a 768 aa, a CRISPR array comprising 10 repeats just downstream of the cas12g gene, and a tracrRNA-encoding region located downstream of the cas12g gene with partial homology to the CRISPR array repeats. The repeat and spacer lengths (36 and 30 bp, respectively) with one of the ten repeats containing 1 bp mutation (Figure 1A). The origin of the spacers in the CRISPR-Cas locus has not been established.

The C-terminus of the candidate gene shows homology to RNA-guided endonuclease TnpB family proteins, which include some CRISPR-associated proteins, such as the type V CRISPR-associated protein C2c8, according to NCBI CDD [31].

### 2.2. Comparison of the Found Cas12g Variant with the Known Homologue

To date, there are several known 3D structures of Cas12g from the hot springs metagenome [29] (PDB codes: 6XMF, 6XMG, 8I16, and 8I3Q) [9,22]. To compare our Cas12g variant with the known ones, we used the 3D structure predicted by AlphaFold [32]. The AlphaFold structure of our Cas12g variant was studied and aligned with UCSF Chimera [33]. The known crystal structure of apo-Cas12g (PDB code: 8l16) was used for comparison [22].

Figure 2 shows that the 3D structure of our Cas12g variant is highly similar to that of the known Cas12g. The overall structure shows a bilobed architecture with the REC and NUC lobes. The WED, RuvC, and Nuc domains are present in the NUC lobe but not the PI domain, confirming that PAM is not necessary for Cas12g to recognize substrates. Our Cas12g variant lacks the key residue substitutions listed in [9,22] that abolish or decrease its enzymatic activity.

Zn^2+^ ion-coordinating the HCCC-type and the CCHC-type zinc finger motifs have been shown to play an important role in the RNA cleavage activity of Cas12g and also influence its thermal stability [9,22]. In turn, the CCCC-type zinc finger influences collateral RNase and single-strand DNase activities [22]. We verified that our Cas12g variant lacks mutations, namely alanine substitutions in the zinc finger motifs [9,22], that critically affect its activity (Figure 2C). Cas12g sequence alignment in Appendix A.

### 2.3. In Vitro Cas12g Characterization

To confirm that the identified CRISPR locus is a functional CRISPR-Cas system, we expressed wild-type Cas12g in *E. coli* with a C-terminal hexahistidine (His)-tag. Protein purification was carried out by metal chelate chromatography and gel filtration (Figure 3A,B). The final protein yield was 1.8 mg.

Then, it was necessary to test the in vitro cleavage activity of the purified Cas12g protein to biochemically characterize the new V-G-type Cas effector. Previously, it was shown that a guide RNA is required for the nuclease to cleave a substrate. We found putative tracrRNA mapping to the region between Cas12g and the CRISPR array (Figure 3C). Given the fact that tracrRNA and crRNA for Cas12g can be fused to form a single guide RNA (sgRNA), we examined whether a sgRNA could be generated for the found Cas12g. Through in vitro transcription, we constructed the sgRNA of Cas12g, which contained a 91-nucleotide tracrRNA at the 5′ end and a 38-nucleotide crRNA at the 3′ end (18-nucleotide repeat and 20-nucleotide spacer sequence).

To determine the ability of the synthesized sgRNA to direct Cas12g to the target locus and cause the cleavage of target molecules, we first decided to test whether a guide RNA and protein complex was formed. To quantify this interaction, we performed the EMSA. The EMSA (electrophoretic mobility shift assay) is a rapid and sensitive method to detect protein–nucleic acid interactions [34,35,36]. Mixtures of protein and RNA in various ratios are analyzed via agarose or polyacrylamide gel electrophoresis. The technique depends on the varying mobility of the nucleic acid, free or bound to protein, in the gel matrix during electrophoresis. After the intensities of shifted and unshifted bands were obtained, we defined the affinity of the interaction of protein and gRNA. Thus, we were able to select optimal conditions for further testing of the found Cas12g (Figure 3D).

Previous studies have shown that Cas12g is capable of cis-cleavage (on-target cleavage of target molecule) and trans-cleavage (promiscuous cleavage of collateral molecule) activities [9,22,29]. To functionally characterize the cis-cleavage activity of the new type V-G Cas effector, we carried out in vitro cleavage reactions. The 8N RNA PAM library (fragment size ~151 nt) obtained by in vitro transcription was incubated with sgRNA and Cas12g (Figure 3E). As expected, cleavage products were observed, represented by major bands of ~100 and ~51 nt. For control mixtures containing neither sgRNA nor Cas12g, no cleavage products were detected, indicating PAM-independent efficient targeting and cleavage of the 8N RNA PAM library of the engineered sgRNA. It can also be observed that upon binding to the target RNA, trans-cleavage activity was activated to nonspecifically cleave surrounding RNA molecules. These findings are consistent with previous reports (Figure 3F).

To confirm that Cas12g is also active on single-stranded DNA, three replicates were carried out in the presence of a DNA probe labeled with a fluorophore and a quencher at its 5′ and 3′ ends, respectively (Figure 4). The rapid and sensitive detection of nucleic acids can have significant applications in disease diagnosis, monitoring, epidemiology, and laboratory tasks [37]. One promising approach involves using Cas12g, which can be programmed with crRNA to trigger programmed cell death or degrade labeled RNA, enabling the detection of specific RNA in vivo or in vitro. This capability for fast and accurate nucleic acid detection could prove to be invaluable in various fields [38].

## 3. Materials and Methods

### 3.1. CRISPR-Cas System Identification and Bioinformatics Analysis

The search of the gene coding for Cas protein was performed in metagenome assemblies of thermophilic microbial communities from Karmadon gorge hot springs, North Ossetia (Russia). Sediments and microbial mats were collected into plastic tubes filled with RNAlater (ThermoFischer Scientific, Waltham, MA, USA) and kept at 4 °C. DNA was extracted using the Qiagen PowerLyzer PowerSoil kit (Qiagen, Germany) according to the manufacturer’s instructions without any modifications, as was reported previously [30]. DNA sequencing libraries were prepared using the NEBNext^®^ Ultra™ II DNA Library Prep Kit (New England Biolabs, Ipswich, MA, USA) according to the manufacturer’s instructions. DNA libraries were sequenced on NovaSeq 6000 (Illumina, San Diego, CA, USA) using the NovaSeq 6000 S2 Reagent Kit (300 cycles) (Illumina, San Diego, CA, USA). The quality of sequencing reads was assessed with FastQC (https://www.bioinformatics.babraham.ac.uk/projects/fastqc/ (accessed on 19 June 2023)). Low-quality regions (trimq = 18), adapter sequences, and poly-G regions of reads were trimmed off using BBDuk ver. 38.90 (https://sourceforge.net/projects/bbmap/ (accessed on 19 June 2023)). Reads were assembled into contigs using SPAdes [39] ver. 3.14.1 and Megahit [40] ver. 1.2.9. CRISPR-Cas loci were searched using CRISPRCasTyper ver. 1.6.1 [41]. Regions containing tracrRNA were defined by a homology search of DR sequences using NCBI blastn [42] in blastn mode with an e-value of 1 × 10^−^³. The secondary RNA structure of the found regions was predicted using the RNAfold web server [43]. Metagenome binning was performed using MetaWrap ver. 1.3 [44] with default settings. The taxonomic classification of metagenomic bins was performed using GTDB-tk [45]. When contig containing CRISPR-Cas loci was not related to any bin, its taxonomy was predicted by searching the NCBI nr database protein for coding sequences in the range of 20,000 b.p. near the CRISPR-Cas locus using NCBI blastp [46] with an e-value of 1 × 10^−6^. The search for homological spacers and DR was conducted using NCBI blastn [46] with an e-value of 1e-6 in blastn mode against NCBI nr and the RefSeq Genome Database. The search of conserved domains was conducted at NCBI CDD [31] with default settings.

### 3.2. Cas12g Structure Analysis and Visualization

The prediction of the 3D structure of Cas12g was performed using AlphaFold [32,47]. The structures were aligned and analyzed using UCSF Chimera [33].

### 3.3. Phylogenetics

The Cas12g homolog was discovered using NCBI BLAST [46] with default parameters against the NCBI nr protein database. The found that sequences were aligned using MAFFT ver. 7.520 [48] in L-INS-i mode. A phylogenetic tree was constructed using IQ-TREE version 2.2.2.7 [49]. The best-fit model WAG+F+I+G4 was chosen according to BIC with ModelFinder, and ultrafast bootstrapping was carried out to assess the branch support values across the tree. The phylogenetic tree was visualized on the iTOL web server [50].

### 3.4. Bacterial Strains and Growth Conditions

The bacterial strain used for the cloning and propagation of plasmids in the current study is *E. coli* Top10. For protein expression, *E. coli* Rosetta (DE3) was used. The *E. coli* strains were routinely cultured at 37 °C and 220 rpm in Luria Bertani medium (LB). Plasmids were maintained with kanamycin (50 mg/mL) and/or chloramphenicol (35 mg/mL), as needed. Liquid media were supplemented with IPTG as specified.

### 3.5. Plasmids Construction

All oligonucleotides were designed in SnapGene software and ordered from Lumiprobe (Russia). All enzymes were purchased from (NEB) and used according to the manufacturer’s protocols, unless otherwise specified. The isolation of plasmids was performed with the Monarch^®^ Plasmid Miniprep Kit (NEB). DNA gel extraction and purification were performed with the QIAquick Gel Extraction Kit (Qiagen, Germany). Sequencing of the resulting DNA constructs was performed by the Sanger method (Prism 3730xl, Applied Biosystems, Carlsbad, CA, USA). All the sequences are shown in the Appendix A.

#### 3.5.1. Recombinant Plasmid Cloning

The Cas12g encoding gene was PCR-amplified using Q5 High-Fidelity 2X Master Mix and primers containing NotI and NdeI restriction sites from 4135 specimens of the extracted environmental DNA [30]. The amplified PCR product was analyzed on 1% agarose gel, and gel was extracted for further cloning. The recombinant plasmid for Cas12g protein was cloned using NotI/NdeI restriction digestion and ligation of the pET30 vector. Competent *E. coli* cells were transformed with a ligase mix and incubated on LB agar plates supplemented with kanamycin. The constructed plasmid was extracted with the Monarch^®^ Plasmid Miniprep Kit according to protocol and confirmed by Sanger sequencing.

#### 3.5.2. PAM Library Construction

The 8N randomized PAM plasmid library was constructed by analogy with the one we previously described in [51]. Briefly, the library was constructed using two M13-flanked synthesized oligonucleotides 100 bp in length consisting of 8 randomized nucleotides. PCR fragments were analyzed and extracted from 1% agarose gel. Purified and digested fragments were cloned into the EagI/PvuII digested pBR322 vector. Competent cells were transformed with a ligase mix, and after 16 h of incubation, ~180,000 colonies were counted. Plasmid library DNA was extracted using the GenElute HP Plasmid Maxiprep Kit (Sigma, Merck KGaA, Darmstadt, Germany). To test the RNA-targeting nuclease, the T7 RNAP promoter was added by PCR amplification using HF Phusion polymerase. PCR fragments were analyzed in 2% agarose gel and then extracted.

#### 3.5.3. gRNA Encoding Plasmid Cloning

The plasmid encoding the template for gRNA synthesis was obtained by sequentially incorporating fragments into the pUC119 scaffold. Initially, two oligonucleotides containing the tracrRNA and T7 RNAP promoter sequences were annealed, and PCR was extended to produce a blunt-ended product for further ligation into HincII-digested pUC119. Then, a BbsI/EcoRI-annealed fragment containing a direct repeat sequence and a spacer was inserted into the resulting construct. The sequence of the resulting plasmid was confirmed by Sanger sequencing.

### 3.6. Expression and Purification of Cas12g Proteins

The C-terminally His_6_-tagged Cas12g protein (the sequence is shown in Appendix A) was purified from *E. coli* Rosetta (DE3) cells transformed with appropriate plasmids. A total of 0.4 L of LB supplemented with 34 μg/mL and 100 μg/mL kanamycin was inoculated with cells from freshly transformed cells. The *E. coli* cells were grown at 37 °C until the OD_600_ reached 0.7. The protein expression was then induced by the addition of 0.3 mM IPTG. The cells were further cultured for 20 h at 18 °C. Cells were harvested by centrifugation and resuspended in buffer A (40 mM Tris-HCl pH, 7.8, 400 mM NaCl, 5 mM imidazole, 0.2% Triton-X100, 1 mM DTT, 1 mM phenylmethylsulfonyl fluoride (PMSF)) and then disrupted by sonication Branson Ultra. Cleared lysates were obtained by centrifugation at 28,000× *g* for 30 min at 4 °C in order to clarify the lysate and applied onto a 1 mL chelating Ni-NTA HisTrap HP (GE Healthcare) equilibrated with buffer B (40 mM Tris-HCl pH 7.8; 400 mM NaCl; 5 mM imidazole; 0.1% Triton-X100). The column was washed with 10 mL buffer B, and the protein was eluted with buffer E (40 mM Tris-HCl pH 7.8; 400 mM NaCl; 300 mM imidazole; 5% glycerol). Protein fractions were pooled and transferred to the final buffer (50 mM Tris-HCl pH 8.0, 500 mM NaCl, 5 mM β-ME, 5% glycerol). Protein fractions during protein isolation were run on 12% denaturing PAGE, followed by Coomassie G-250 staining. The protein concentration was determined by the Qubit protein assay (Thermo Fisher, Carlsbad, CA, USA) and stored at −80 °C until use.

### 3.7. In Vitro RNA Preparation

The DNA template for sgRNA transcription containing a T7 RNAP promoter was obtained by PCR amplification with M13 primers. The PCR fragment was run on agarose gel and then gel-purified. The resulting PCR fragment was cut with the MseI restriction enzyme and purified prior to transcription. *MseI* digestion leaves a 5′-overhanging end suitable for in vitro transcription. Guide RNA synthesis was obtained with the HiScribe T7 High Yield RNA Synthesis Kit. The reaction mix consisted of 1x reaction buffer, 1 μg of DNA template, rNTP mix with a final concentration of 10 mM each, T7 RNA Polymerase Mix, and 1 μL of RNasine and nuclease-free water to a final volume of 40 μL. The reaction was incubated at 37 °C overnight. To remove template DNA, the resulting mixture was incubated with DNase I in 1xDNaseI buffer for 30 min at 37 °C. RNA extraction was performed with the TRIZOL-chloroform method and ethanol precipitation. Finally, the gRNA pellet was dissolved in RNase-free water. The obtained RNAs were confirmed by 8% UREA-TBE (8M urea) polyacrylamide gel electrophoresis stained with EtBr. The RNA concentration was measured on Qubit 3.0 using a Qubit RNA HS Assay Kit (Thermo Scientific, Carlsbad, CA, USA).

### 3.8. Electrophoretic Mobility Shift Assay (EMSA)

Binding assays were performed by incubating 0, 0.2, 0.5, 1.0, 2.0, 3.0, and 4.0 µg fresh working stock dilutions of Cas12g with 1 µg of gRNA for 30 min (not longer, as complexes might dissociate) at 37 °C in an equilibration reaction buffer (10 mM Tris-HCl, 10 mM MgCl_2_, 50 mM NaCl, 1 mM DTT, pH 7.5). Reaction products were run on 1% agarose gel for 4 h to check the combining efficiency, and electrophoresis was performed in a 1 × TBE buffer. Gels were imaged with the Gel Doc EZ System (Bio-Rad, https://www.bio-rad.com/, accessed on 19 June 2023). Bands were quantified using ImageLab software on the assumption of the percentage of bend intensity.

### 3.9. In Vitro Cleavage Assay

The cleavage activity of the Cas12g protein was measured by in vitro RNA cleavage experiments. The Cas12g–guide RNA complex was prepared by mixing purified Cas12g protein with its cognate guide RNA (crRNA and tracrRNA) at 37 °C for 15 min. The pre-assembled Cas12g–crRNA complex was mixed with the target RNA (at 1:3:30 target/Cas12g/gRNA ratio) and then incubated at 37 °C for 60 min in 1xNebuffer 2. The reaction was stopped by the addition of Proteinase K (40 ng). The reaction products were incubated at 95 °C for 5 min with RNA loading dye buffer for denaturing and then fractionated by electrophoresis using 10% TBE-Urea-PAGE gel and analyzed on the Gel Doc EZ System (Bio-Rad). In vitro cleavage experiments were performed at least three times.

### 3.10. Collateral Activity

The ssDNA target labeled at the 5’ end with the ROX fluorophore was used to determine the collateral activity of the nuclease of nontarget ssDNA. The pre-assembled Cas12g–sgRNA complex (1:2 molar ratio) was mixed with the target RNA. The final concentration of labeled substrate was 100nM. Target cleavage assays were set up in the optimized cleavage buffer (10 mM Tris-HCl, 10 mM MgCl2, 50 mM NaCl, 1 mM DTT, pH 7.5) at 37 °C for 30 min. Reactions were first treated with an RNase cocktail with incubation at 37 °C for 15 min. Next, they were treated with Proteinase K and incubated at 37 °C for 15 min. To detect ssDNA cleavage products, the reactions were analyzed with a Varioskan LUX multimode microplate reader (ThermoFisher). Excitation filter: 575 nm; emission filter: 602 nm.

## 4. Conclusions

The development of RNA-targeting CRISPR-Cas systems has opened up a promising avenue for advancing research and therapeutic applications. With the ability to target RNA, researchers can now manipulate gene expression at the post-transcriptional level, allowing for greater precision in gene editing and regulation [52]. Cas12g, a recently found RNA-guided nuclease, was identified by analyzing the metagenomes from promising thermophilic microbial communities. Unlike other type V Cas effectors, which mainly target DNA substrates, Cas12g specifically targets RNA substrates [53,54,55,56]. Based on the 3D structure predicted by AlphaFold, our Cas12g nuclease contains a single RuvC domain that can cleave the ssRNA molecule and collateral ssDNA, as was demonstrated by in vitro experiments.

Although CRISPR-Cas12g systems are currently under extensive research, with Cryo-EM structures already obtained and the substrate cleavage mechanism elucidated, along with potential methods for repurposing Cas12g, there are still some challenges to overcome before RNA-targeting CRISPR-Cas systems can be widely adopted. One of the main challenges is off-target activity. While CRISPR-Cas systems are known for their high specificity, there is still a risk of unintended cleavage of nontargeted RNA molecules. This can lead to unwanted side effects and potentially harmful consequences. Researchers are currently working on improving the specificity of RNA-targeting CRISPR-Cas systems by optimizing the guide RNA design and delivery methods.

Delivery is another issue that needs to be addressed. RNA-targeting CRISPR-Cas systems require the delivery of both the Cas protein and the guide RNA into the target cells, so the same is true for all Cas systems. This can be challenging, especially for certain cell types or tissues. Researchers are exploring different delivery methods, such as viral vectors or lipid nanoparticles, to improve the efficiency and specificity of RNA-targeting CRISPR-Cas systems. Despite these challenges, the potential clinical applications of RNA-targeting CRISPR-Cas systems are vast.

The multifaceted nature of Cas12g, which possesses both collateral RNase and single-stranded DNase activities, makes it an ideal tool for advancing biotechnology and biomedical research. The methods outlined in this study can be applied to other Cas orthologs and variants, leading to a better understanding of the CRISPR-Cas system and its enzymatic activities. Continued development in this field will undoubtedly lead to advancements in both research and therapeutic applications. Ultimately, the search for new genome editing tools and the study of their properties will remove barriers to research in this field, leading to new discoveries and innovations in biotechnology and medicine. RNA-targeting CRISPR-Cas systems have the potential to revolutionize the way we treat genetic disorders and infectious diseases, and with continued research and development, we may be able to unlock their full potential.

## Figures and Tables

**Figure 1 ijms-24-17105-f001:**
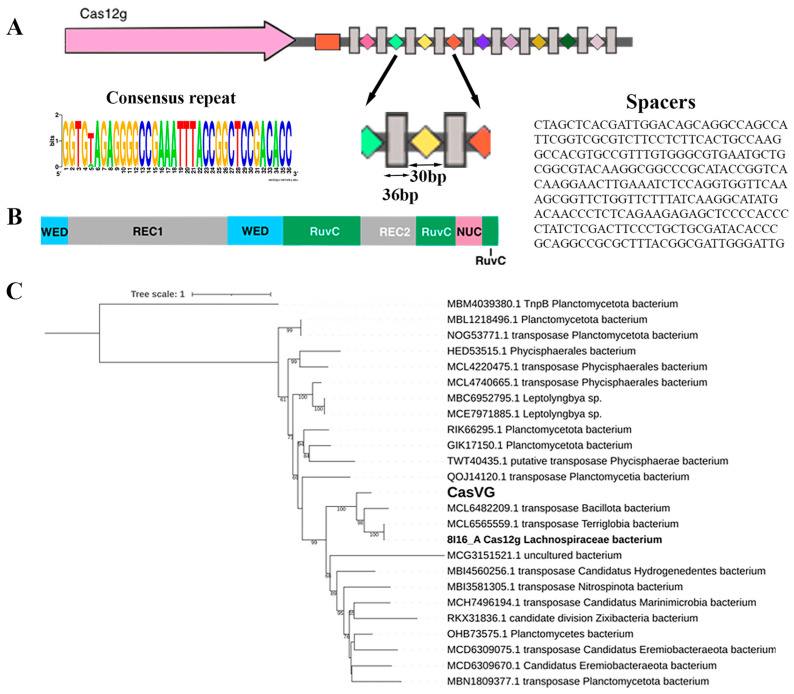
CRISPR-Cas type V locus: (**A**) organization of the Cas12g locus. Direct repeats (DRs) are shown by gray rectangles, and spacers are indicated by rectangles of different colors. The tracrRNA coding sequence is shown in the orange box. The *cas* gene is labeled; (**B**) schematic representation of Cas12g domain organization; (**C**) maximum likelihood phylogenetic tree of Cas12g nuclease. Bootstrap values below 50 were not shown. MBM4039380.1 is an outgroup.

**Figure 2 ijms-24-17105-f002:**
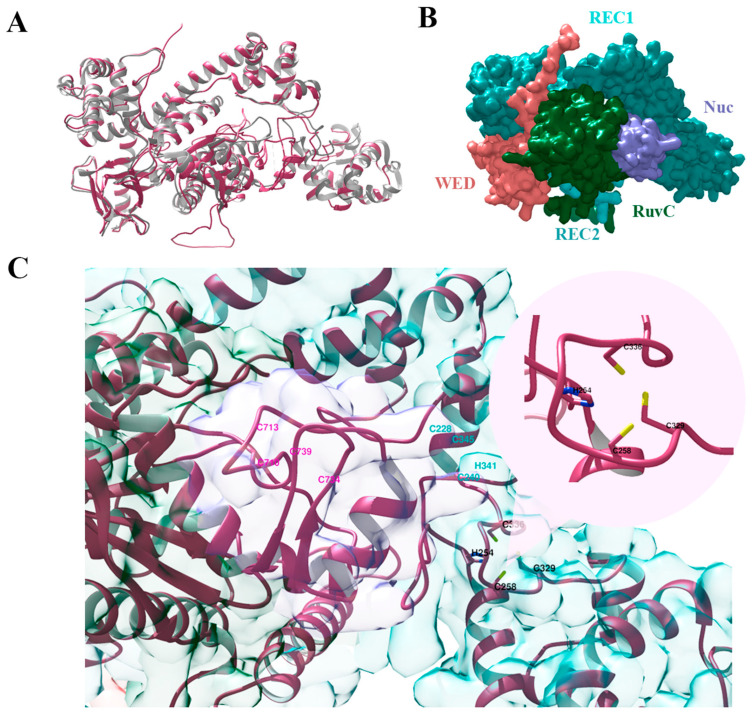
Three-dimensional structures of Cas12g protein: (**A**)—alignment of the predicted structure of Cas12g protein (pink) and the known Cas12g (grey) (PDB code: 8l16); (**B**)—the surface representation of Cas12g protein; (**C**)—three types of zinc finger motifs: the HCCC-type (black), the CCHC-type (cyan), and the CCCC-type (pink) zinc fingers.

**Figure 3 ijms-24-17105-f003:**
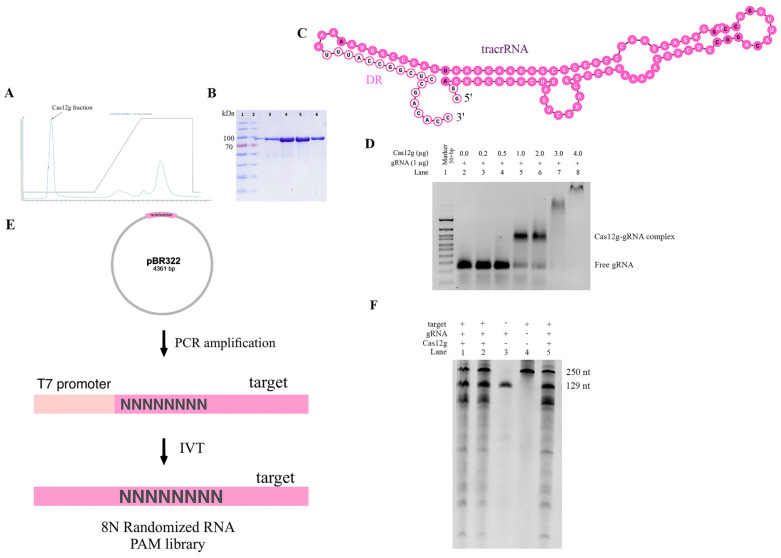
In vitro characteristics of new Cas12g nuclease: (**A**) chromatogram showing elution of Cas12g protein; (**B**) SDS-PAGE (sodium dodecyl sulfate polyacrylamide gel electrophoresis) analysis of affinity enriched Cas12g fraction (Ni-NTA Elution); (**C**) in silico co-folding of DR Cas12g and putative tracrRNA. The DR sequence is colored bright pink, and the tracrRNA sequence is colored pale pink. The nucleotides highlighted in bold denote a lower probability of being paired or unpaired compared to other nucleotides; (**D**) EMSA assay; (**E**) RNA PAM library preparation; (**F**) in vitro cleavage activity of Cas12g nuclease.

**Figure 4 ijms-24-17105-f004:**
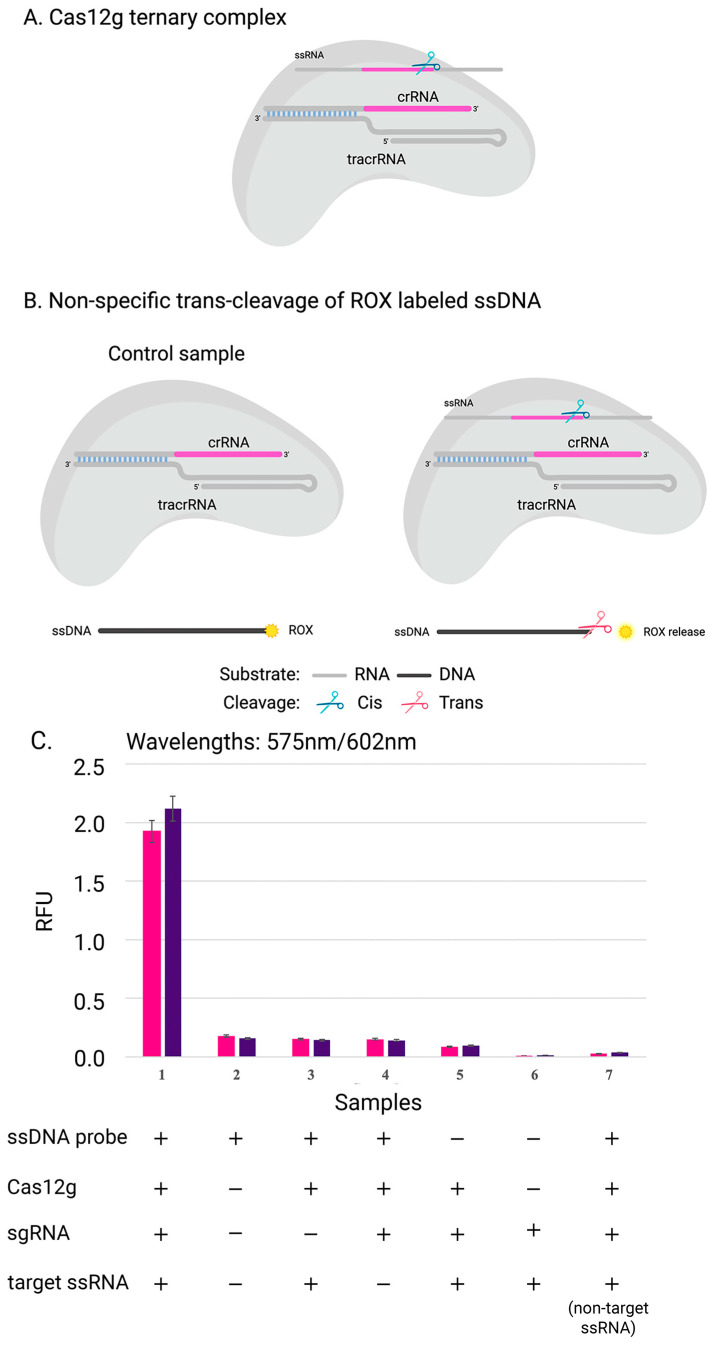
Scheme of experiment: (**A**) complex of Cas12g with guide RNA; (**B**) the collateral activity of nuclease Cas12g on ssDNA target. (**C**) The cleavage efficiency of Cas12g nuclease on distinct ssRNA targets: results of experiments with two sgRNA-target ssRNA pairs (pink and purple) are shown. The data show the mean value from the results of three independent experiments.

## Data Availability

All data are presented in the manuscript or in the Appendix A.

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
