# Peer review of "A New RNA-Dependent Cas12g Nuclease"

_ijms, 2023, doi:10.3390/ijms242317105_

Round 1
Reviewer 1 Report
Comments and Suggestions for Authors
The work by Gunitseva described a newly discovered Cas12g protein and reported the cis- and trans-cleavage activity of this nuclease. Overall, the discovery is of sufficient interest to the scientific community. However, the experimental design did not sufficiently justify the statements in the study. Therefore, additional experiments should be performed for a more thorough analysis.
Major
1. For Fig.3E-F, it appeared that the authors only tested one pair of gRNA and substrate. It is recommended that the authors use more than one pair of gRNA and substrate RNA of different sequences for assessing the cis-cleavage activity.
2. It is suggested that authors design experiments to reveal the PAM sequences of this Cas12g to maximize the utility of the system for readership.
3. For Fig.4C, authors are suggested to repeat the experiments with different gRNA (and corresponding substrate RNA) to exclude the possibility that the observed signal was due to specific cleavage rather than trans-cleavage activity.
4. To maximize the utility of the discovery, the authors are encouraged to examine whether RPA or LAMP could be combined with the Cas12g-based diagnosis for improved signal output.
Minor
1. The figures presented in the article should be cited in sequential order. It seems to the reviewer that many figures are not cited in the text in sequential order, such as Fig.3A-3D.
2. Fig.3B and 3F, the molecular weight of bands of interest should be indicated.
3. What is zond in Fig.4C? Is that the ssDNA reporter?
Reviewer 2 Report
Comments and Suggestions for Authors
This study introduces the CRISPR-Cas12g system, an adaptive immune system with the ability to control gene expression and influence RNA molecules. It specifically highlights a newly identified Cas12g effector protein from thermophilic microbial communities’ metagenomes. This protein demonstrates the ability to cleave both single-stranded DNA (ssDNA) and single-stranded RNA (ssRNA) molecules, with both cis- and trans-cleavage activities. The text suggests that Cas12g nucleases have the potential to become widely used tools for RNA editing, with significant implications for biotechnology and biomedical research. Exploring new genome editing tools is expected to advance research in this domain, ultimately leading to fresh discoveries and innovations in biotechnology and medicine.
In summary, while the abstract introduces an intriguing topic and results provide important findings, however, it lacks the depth and specificity required for a scientific or informative piece. It should include more detail about the research, citations, and a more balanced discussion of the potential and challenges of Cas12g nucleases in genome editing.
Authors can address some relevant queries:
- The abstract provides a brief overview of the CRISPR-Cas12g system and its potential applications, but it lacks crucial details. It doesn't delve into the specifics of the research conducted, which limits its informativeness.
- Again in the abstract authors say "they are able to regulate gene expression and affect RNA molecules" and "we have indicated cis- and trans cleavage activities" lack specificity. It doesn't describe the mechanisms or experimental results that support these claims.
- The conclusions and abstract lacks statistical values to specific studies or experiments. It's important to provide data in support of scientific claims to establish credibility and allow readers to explore the topic further.
- Overly Optimistic Language: The text presents the potential of Cas12g nucleases in an overly optimistic manner, suggesting they "may become a widely used tool" without concrete evidence or discussion of current limitations and challenges.
- How does this work relate to previous studies in the field of CRISPR-Cas systems or genome editing? Providing context is essential for readers to understand the significance of the research.
- Authors could improve scientific literature for discussion and introductions, please see: PMID: 34975291 and PMID: 36843874.
- Figure 3 A and C; and Figure 4: quality needs to be improved.
Comments on the Quality of English Language
- Grammar and Syntax: There are grammatical issues, such as "an widely used tool" instead of "a widely used tool," which can affect the text's professionalism and clarity. Similar misusage of language could be avoided.
- Clarity needs to be improved. The text is somewhat disjointed and could benefit from a more structured flow of ideas. It doesn't transition smoothly between introducing the Cas12g system, describing the new ortholog, and discussing potential applications
Reviewer 3 Report
Comments and Suggestions for Authors
In the present study, the authors identified a new type of Cas12g nuclease from thermophilic microbial community metagenomes. This Cas12g is an effector protein homolog to previously identified Cas12g and belongs to the type V of the CRISPR-Cas system. Using AlphaFold, the candidate Cas12g was found to predict similar 3D structures to the previously crystallized Cas12g structure from Lachnospiraceae bacterium. The researchers purified the candidate Cas12g protein and confirmed its RNA targeting cleavage (cis-cleavage) and collateral RNA and ssDNA cleavage (trans-cleavage) activity, which mirrors that of previously identified Cas12g effector proteins. CRISPR technology is crucial for gene-editing and regulation in biotechnology and biomedical applications and has made significant progress in the last decade. This work aims to expand the toolkit of CRISPR-Cas effector proteins through metagenomic analysis in thermophilic microbial communities, which is of importance. However, there are several key weaknesses in this study.
Broadly, three major issues are evident in this work. Firstly, the authors do not clearly delineate what distinguishes this newly identified Cas12g from its predecessors, aside from its sequence, particularly considering its source in a unique environment. Secondly, the authors claim that Cas12g shows great potential for RNA editing but also acknowledge its collateral (non-specific) RNase activity, which would significantly hinder its in vivo RNA editing application. This has been reported from previous Cas12g studies, and this new Cas12g does not show advantage/superior. Thirdly, the presentation of the work requires substantial improvement, as detailed below.
The title should be revised to "A New RNA-dependent Cas12g Nuclease."
Please introduce the definitions of cis- and trans-cleavage to enhance reader understanding.
Discrepancies between the discussed RNA-targeting system and referenced material about DNA-targeting systems need clarification (e.g., Line 52 and 53 referencing RNA but citing a Cas9 DNA targeting system).
On line 79, the statement "we investigate previously undescribed Cas12g nuclease" should include "a" before "previously" to avoid confusion about whether the authors discovered Cas12g. This clarification is essential, as the same name is used for different Cas12g from various species. Additional clarifications are recommended regarding the distinction between previous findings on Cas12g and the work conducted in this study. Moreover, please provide a protein sequence alignment between these different Cas12g.
Figure 3A-D is missing in the main text and should be placed adjacent to the corresponding text description.
In Figure 3D, please label the 5' and 3' ends of RNA and make the distinction between bright pink and pale pink more apparent. Clarify why some of the nucleotides are bolded.
Figure 3E requires labels for marker size and different bands in the gel (e.g., free gRNA vs. cas12-bound gRNA).
Figure 3F needs labeled bands in the gel and clarification regarding the differences between lane 1, 2, and 5.
Figure 4 necessitates an elaboration of its contents, the detection mechanism/model, and how the data proves or disproves the working hypothesis. Better explanations, particularly regarding the x-axis in Figure 4C, are required. Furthermore, it is noted that this assay directly reads the Cas12g trans-cleavage activity on ssDNA, not the targeting RNA cleavage activity.
Other minor comments:
Ref 22 is incorrect.
Line 51: Missing reference(s).
Line 58: Missing reference(s).
Line 109: Spell out "DR" since it has not been introduced in this paper.
Line 116: The "I" in PDB ID "8I16" looks different from that in "8I3Q."
Line 120: Add "predicted" before "3D structure of…"
Lines 128-129: The sentence is unclear. Do the authors intend to say that the new Cas12g is similar to previous Cas12g with regard to the zinc finger motif and activity?
Line 132: The term "well-known" is inaccurate. Please note that this is a structure published only two months ago. Additionally, annotation of the PDB ID in the figure legend is suggested.
Lines 134-135: The corresponding text in the figure is very small and blurry.
Line 315: Provide information about the concentration and ratio of ssDNA in this assay.
Comments on the Quality of English Language
The English in this paper requires extensive revision. The authors frequently omit the use of "a" before certain statements, causing confusion and potential misunderstanding. It's crucial to put in additional effort to differentiate between previously published work on Cas12g and the findings of this study.
Round 2
Reviewer 1 Report
Comments and Suggestions for Authors
In this revision, the authors have adequately addressed most of the reviewer's concern except one. The reviewer still has concerns on Fig. 4C.
As the conclusion drawn from this figure is the "non-specific cleavage or trans-activity on ssDNA", experimental evidence must be provided that the cleavage is not dependent on specific sequence of ssDNA, sgRNA or substrate RNA. In other words, the authors should at least test one variation (ssDNA, sgRNA or substrate) to conclude that this cleavage is real "non-specifc trans-cleavage".
Comments on the Quality of English LanguageEnglish is fine.
Reviewer 3 Report
Comments and Suggestions for Authors
The authors failed to address the first two major issues I raised regarding this study, focusing predominantly on resolving the third point. While I acknowledge that there may be limited options available at this juncture, I suggest that they incorporate additional comments in the work to enhance its rigor. While the authors have successfully addressed most of my concerns related to the third point, I still contend that it is crucial to include sequence alignment for both the previously identified Cas12g and the newly discovered Cas12g in this study, despite having only two sequences.While the authors have presented 3D structure comparisons between the previous Cas12g and the AlphaFold-predicted new Cas12g, it's important to note that the latter is based on AI-predicted structures and lacks validation, unlike the objectivity of sequence differences. Sequences can sometimes differ without corresponding structural disparities, and conversely, there can be significant structural differences with only a few mutations. It remains unclear in this specific case whether the observed difference in Figure 2A is attributed to sequence divergence. Therefore, I recommend including the sequence alignment as a supplementary figure for clarification.
Furthermore, in line 184, it is unclear which specific "three independent in vitro experiments" the authors are referring to, especially considering there are six groups in Figure 4C. Clarity is needed to specify the exact experiments being discussed to avoid any potential ambiguity.
Round 3
Reviewer 1 Report
Comments and Suggestions for Authors
The authors have partly addressed my concerns by providing experimental evidence that a non-specific substrate RNA will not induce the trans-cleavage activity of Cas upon recognition of a certain crRNA. An experiment showing that a second substrate RNA can be recognized by a second crRNA to trigger the trans-cleavage will be more convincing for the general conclusion.
